

# Mahalanobis distances and ecological niche modelling: correcting a chi-squared probability error

Thomas R. Etherington

Manaaki Whenua—Landcare Research, Lincoln, New Zealand

## ABSTRACT

The Mahalanobis distance is a statistical technique that can be used to measure how distant a point is from the centre of a multivariate normal distribution. By measuring Mahalanobis distances in environmental space ecologists have also used the technique to model: ecological niches, habitat suitability, species distributions, and resource selection functions. Unfortunately, the original description of the Mahalanobis distance technique for ecological modelling contained an error describing how Mahalanobis distances could be converted into probabilities using a chi-squared distribution. This error has been repeated in the literature, and is present in popular modelling software. In the hope of correcting this error to maximise the potential application of the Mahalanobis distance technique within the ecological modelling community, I explain how Mahalanobis distances are calculated, and through a virtual ecology experiment demonstrate how to correctly produce probabilities and discuss the implications of the error for previous Mahalanobis distance studies.

## INTRODUCTION

The Mahalanobis distance (*Mahalanobis, 1936*) is a statistical technique that can be used to measure how distant a point is from the centre of a multivariate normal distribution. Consider a data matrix $\mathbf{A}$ with $m$ rows of observations and $n$ columns of measured variables.

$$\mathbf{A} = \begin{bmatrix} x_{11} & x_{12} & x_{13} & \dots & x_{1n} \\ x_{21} & x_{22} & x_{23} & \dots & x_{2n} \\ x_{31} & x_{32} & x_{33} & \dots & x_{3n} \\ \vdots & \vdots & \vdots & \ddots & \vdots \\ x_{m1} & x_{m2} & x_{m3} & \dots & x_{mn} \end{bmatrix}.$$

The Mahalanobis distance $D^2$ for each observation vector $\mathbf{x}_m = [x_{m1}, x_{m2}, x_{m3}, \dots, x_{mn}]$ is calculated as a function of an $n$-dimensional vector $\bar{\mathbf{x}}$ containing the means for each column of variables, and a variance–covariance matrix $\mathbf{S}$ of dimensions $n \times n$ that contains variances for each column along the main diagonal and pair-wise column covariances

Corresponding author
Thomas R. Etherington,
etheringtont@landcareresearch.co.nz

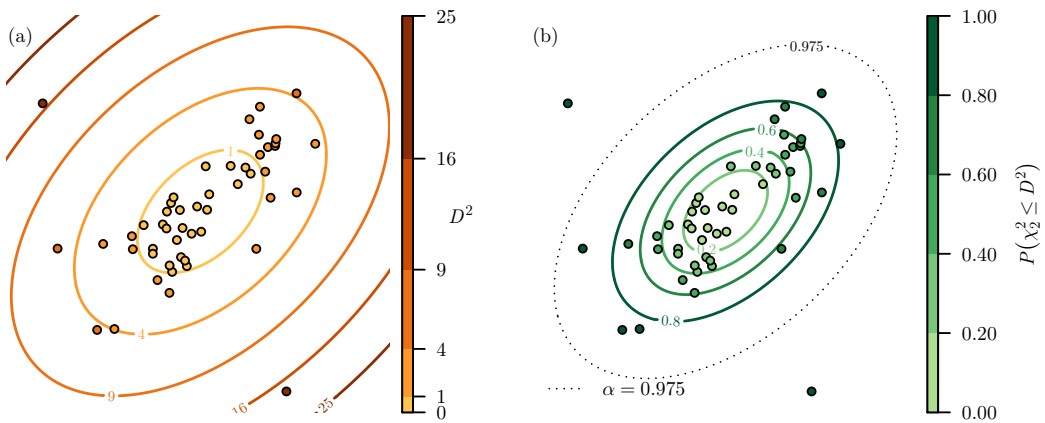

**Figure 1 Two-dimensional example of Mahalanobis distance.** (A) Given a set of points distributed in two-dimensional space, the Mahalanobis distances ($D^2$) for each point can be calculated. (B) The $D^2$ values can be transformed into probabilities using a chi-squared cumulative probability distribution. This highlights that there are two points points that have a very high probability of not belonging to the distribution and could be classified as outliers when $\alpha = 0.975$ as they are beyond that probability threshold.

values elsewhere (*Manly, 2005*).

$$D^2(\mathbf{x}_m) = (\mathbf{x}_m - \bar{\mathbf{x}})^{\mathrm{T}} \mathbf{S}^{-1} (\mathbf{x}_m - \bar{\mathbf{x}}). \tag{1}$$

When applied to $m = 50$ points in $n = 2$ dimensions, the calculated $D^2$ values follow a characteristic elliptical pattern with $D^2$ radiating out from the central location of the distribution (Fig. 1A).

While $D^2$ can be calculated for any $n$-dimensions, the values of $D^2$ are not comparable when $n$ varies, as $D^2$ increases as $n$ increases (Figs. 2A–2C). However, the $\bar{\mathbf{x}}$ and $\mathbf{S}^{-1}$ used to calculate $D^2$ (Eq. (1)) transform the values from each $n$ into independent standard normal distributions, with $\bar{\mathbf{x}}$ centering and $\mathbf{S}^{-1}$ scaling and rotating each variable distribution. This means that $D^2$ is essentially the sum of $n$ independent standard normal variables, and as such follows a chi-squared distribution with degrees of freedom equal to the number of dimensions $n$ (*Manly, 2005*).

For a chi-squared random variable with $n$ degrees of freedom denoted as $\chi_n^2$, the probability density function $f$ of $\chi_n^2$ when $x \geq 0$ is

$$f_{\chi_n^2}(x) = \frac{1}{2^{n/2}\Gamma(n/2)} e^{-x/2} x^{(n/2)-1} \tag{2}$$

where $\Gamma$ is a gamma function (as the chi- squared distribution is actually a special case of the gamma distribution), and the associated cumulative distribution function is $F_{\chi_n^2}(x) = P(\chi_n^2 \leq x)$ (*Johnson, Kotz & Balakrishnan, 1994*). When the variables in **A** are normally distributed, the association between $D^2$ and $f_{\chi_n^2}(x)$ can be clearly seen (Figs. 2A–2C).

By converting $D^2$ into probabilities using $F_{\chi_n^2}(x)$ we can put $D^2$ from any number of dimensions on a common 0–1 scale that indicates the probability $P(\chi_n^2 \leq D^2)$ that a location has a $D^2$ that is greater than that would be expected by chance (Figs. 2D–2F).

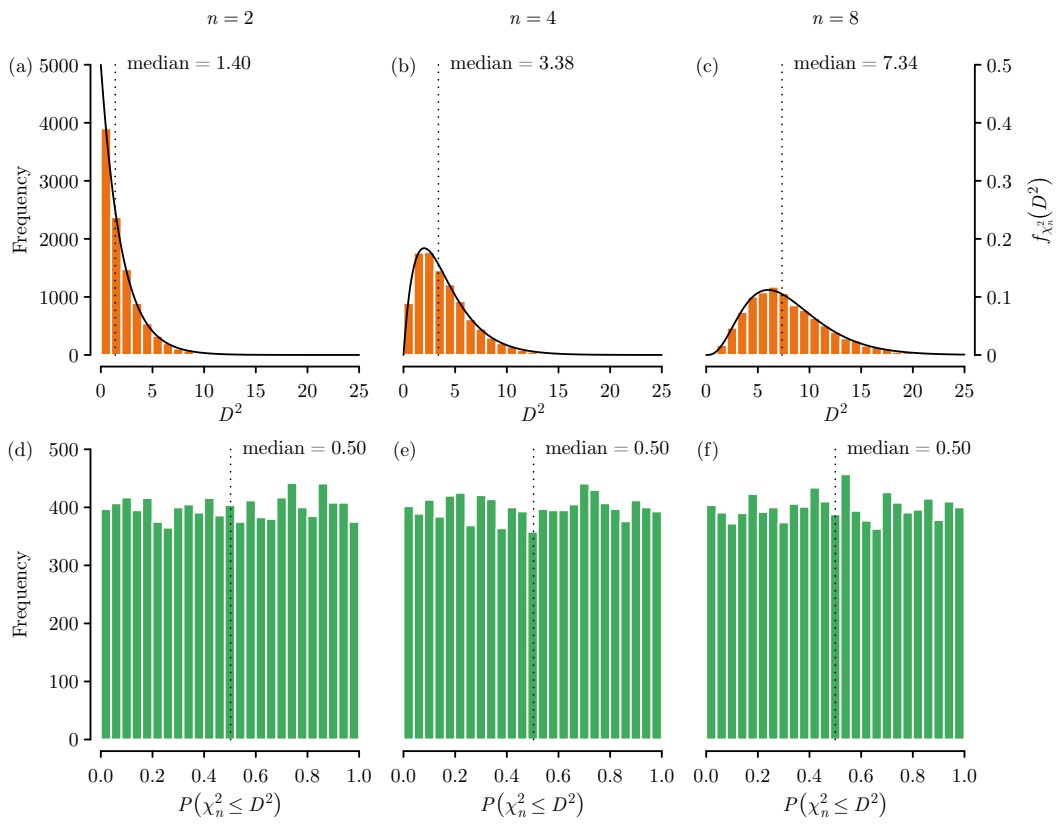

**Figure 2** **Mahalanobis distance ($D^2$) dimensionality effects using data randomly generated from independent standard normal distributions.** We can see that the values of $D^2$ grow following a chi-squared distribution as a function of the number of dimensions (A) $n = 2$, (B) $n = 4$, and (C) $n = 8$. By using a chi-squared cumulative probability distribution the $D^2$ values can be put on a common scale, such that all values range 0–1 and so that statistics such as the median are consistent across dimensions of (D) $n = 2$, (E) $n = 4$, and (F) $n = 8$. The near uniform distribution of the probability values is expected given the underlying data were randomly generated from independent standard normal distributions.

By specifying a significance level $\alpha$ this process is commonly used as an outlier detection method as it is: parameter free, computational efficient, accounts for collinearity between variable dimensions, and is scale independent (*Aggarwal, 2017*). Returning to our earlier example, having transformed the values from $D^2$ to $P(\chi_n^2 \leq D^2)$ we can see that there are two points that are very likely to be outliers, and would be classified as outliers with $\alpha = 0.975$ (Fig. 1B).

The potential of $D^2$ for use in habitat modelling was first identified by *Clark, Dunn & Smith (1993)* and then discussed further in the context of niche modelling by *Farber & Kadmon (2003)*—which is how I will continue the discussion. The premise here is that given a data matrix **A** of $m$ species observations for which various environmental variables $n$ are measured, $D^2$ can be used as a measure of niche suitability from an optimum location in environmental space. Having defined a niche in this way, by measuring the $D^2$ for each location on a landscape a map of niche suitability can then be produced. The key advantages of using $D^2$ over other methods are that the $D^2$ method needs only presence information,

and so does not require either absences or a background definition, and that independence of explanatory variables is not required (*Clark, Dunn & Smith, 1993*; *Farber & Kadmon, 2003*). Studies that have compared Mahalanobis distance to other modelling approaches have also shown that while the optimum method tends to vary by the species in question, the Mahalanobis distance approach performs well against a variety of other presence-only, presence-background, and presence-absence modelling approaches (*Dettmers, Buehler & Bartlett, 2002*; *Johnson & Gillingham, 2005*; *Tsoar et al., 2007*).

While introducing $D^2$ into the ecological modelling domain *Clark, Dunn & Smith (1993)* also highlighted that $D^2$ will follow a chi-squared distribution, and therefore the potential to convert the distances into probabilities. However, as ecological niche models aim to describe probability of belonging to the niche, rather than use the chi-squared cumulative distribution function $F_{\chi_n^2}(x) = P(\chi_n^2 \leq x)$ that has 0 at the optimum, we use the inverse chi-squared cumulative distribution function $F_{\chi_n^2}^{-1}(x) = P(\chi_n^2 > x)$ that has 1 at the optimum. This means that the probabilities $P(\chi_n^2 > D^2)$ indicate locations with a $D^2$ that is less than that would be expected by chance, and hence are more likely to be within the niche.

Unfortunately, when describing the use of a chi-squared distribution to convert $D^2$ into probabilities, *Clark, Dunn & Smith (1993)* p.522 state that "Assuming multivariate normality, Mahalanobis distances are approximately distributed as Chi-square with $n - 1$ degrees of freedom, where $n$ equals the number of habitat characters.", but this is incorrect. $D^2$ values follow a chi-squared distribution with degrees of freedom equal to $n$ (*Manly, 2005*) as has already been clearly shown (Fig. 2). This error has been repeatedly described in the literature (*Knick & Rotenberry, 1998*; *Farber & Kadmon, 2003*; *Hellgren et al., 2007*), and has even permeated into software such as the R package adehabitat (*Calenge, 2006*).

To demonstrate this error and to examine its implications, I present an experiment based on a virtual ecology approach (*Zurell et al., 2010*)—which allows us to examine methodologies in a controlled system uncomplicated by the uncertainties of the real world!

## MATERIALS AND METHODS

The virtual ecology experiment began by defining, and therefore knowing truthfully, the fundamental niche of the imaginary species *Mimbulus mimbletonia* (*Rowling, 2003*). This fundamental niche $N$ was defined using a multivariate normal distribution describing the niche in relation to two environmental variables of temperature and rainfall.

$$N_{(x,y)} = e^{-\frac{1}{2}((x,y)-\mu)^{\mathrm{T}}\Sigma^{-1}((x,y)-\mu)}, \text{where } \mu = \begin{bmatrix} 25 & 100 \end{bmatrix} \Sigma = \begin{bmatrix} 9 & 60 \\ 60 & 625 \end{bmatrix}. \tag{3}$$

A sample of observations was then created by randomly sampling the niche space ranging 15–35 °C of temperature and 0–200 mm of rainfall. At each randomly selected location the species was considered detected using a probability equal to the fundamental niche. This process was continued until a sample of 200 observations was generated.

The values of temperature and rainfall at each of the 200 sampled locations were then used to estimate the fundamental niche using Mahalanobis distances. The $D^2$ was calculated

for each of the sampled locations, and the resulting distribution of $D^2$ was compared against a range of inverse cumulative chi-squared distribution functions based on differing degrees of freedom.

Finally, the $D^2$ values for the samples were converted into probabilities of belonging to the fundamental niche, and these predictions were compared against the known fundamental niche values for the locations. Probabilities were calculated using chi-squared distributions with $n$ and $n-1$ degrees of freedom to examine any differences.

## RESULTS

The fundamental niche defined using the multivariate normal distribution (Eq. (3)) produced an elliptically shape niche with positive correlation between rainfall and temperature (Fig. 3A). The random sampling resulted in a set of 200 samples that followed this elliptical niche pattern, with a greater concentration of samples towards the centre of the niche (Fig. 3B).

The $D^2$ values calculated on the basis of these samples also followed an elliptical pattern (Fig. 3C), and when the calculated $D^2$ values for each sample were plotted against the actual fundamental niche value for each sample, a trend that clearly follows the inverse cumulative chi-squared distribution when $n=2$ can be seen (Fig. 3D). This is as we would expect as in this example the fundamental niche is based on the two environmental variables of temperature and rainfall.

When the $D^2$ values were converted to probabilities using $n=2$ degrees of freedom we see a near-perfect linear fit between the estimated niche suitability and the actual known niche suitability (Fig. 3E). This proves quite clearly that to get a truthful estimate of our known fundamental niche, the probabilities need to be based on an inverse cumulative chi-squared distribution with $n$ degrees of freedom. In contrast, when the $D^2$ values are converted to probabilities using $n-1$ degrees of freedom we see a badly-fitting curvilinear relationship that underestimates niche suitability (Fig. 3F).

## DISCUSSION

The results of the virtual ecology experiment (Fig. 3) clearly shows the erroneous under prediction of niche suitability when using a chi-squared distribution with $n-1$ degrees of freedom. Fortunately most previous $D^2$ studies simply rescaled the $D^2$ into quantiles or ranks of increasing suitability (*Knick & Dyer, 1997*; *Knick & Rotenberry, 1998*; *Johnson & Gillingham, 2005*; *Hellgren et al., 2007*; *Etherington et al., 2009*) or binary classifications based on a threshold (*Farber & Kadmon, 2003*; *Thatcher, Van Manen & Clark, 2006*; *Tsoar et al., 2007*) and so are not affected by this problem. Also, although I could not find any examples of this, those studies that did create chi-square probabilities using $n-1$ degrees of freedom, but then converted to categories based on quantiles or a predictive threshold would not have impacted the conclusions of the study as while the shape of the relationship becomes curved with $n-1$, the trend is still one of monotonic increase, therefore the quantiles would be the same. However, those studies that have created

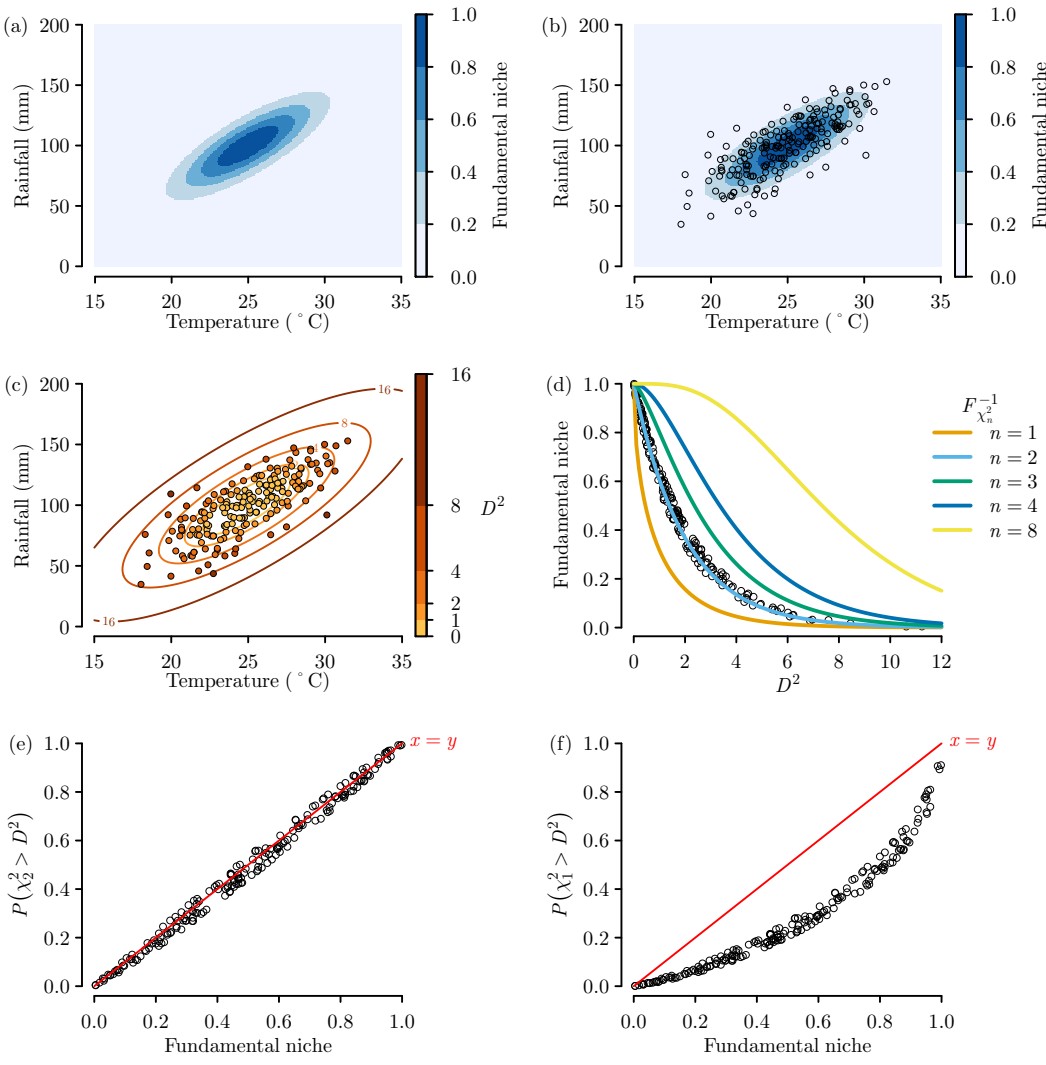

**Figure 3 Virtual ecology experiment to examine the importance of chi-squared distribution degrees of freedom.** (A) The virtual fundamental niche with (B) random sampling across niche space and (C) resulting $D^2$ calculations. (D) The trend of $D^2$ compared against the inverse chi-squared cumulative probability distributions with differing degrees of freedom $n$. The trend for the sampling locations between the actual fundamental niche values and the niche estimates from $D^2$ values converted to probabilities via an inverse cumulative probability distribution with degrees of freedom equal to (E) $n = 2$ and (F) $n = 1$ (or $n = 2 - 1$).

chi-squared probabilities with $n - 1$ as some form of suitability index (*Clark, Dunn & Smith, 1993*) will have underestimated the niche suitability.

## CONCLUSION

Given that $D^2$ values are unitless and unbounded and are not directly comparable for different dimensions, I would argue that anyone using the Mahalanobis distance method should present their results as chi-squared probabilities. This will put Mahalanobis distance

models on a 0–1 scale that enables models based on differing numbers of $n$ to be directly comparable, and is consistent with most other types of ecological niche models that also use a 0–1 scale. As such, it will be very important that the chi-square probabilities are calculated correctly and hopefully the methodological description and experimental evidence presented here will enable that to be achieved.

### Funding
This research was funded by internal investment by Manaaki Whenua—Landcare Research. The funders had no role in study design, data collection and analysis, decision to publish, or preparation of the manuscript.

### Grant Disclosures
The following grant information was disclosed by the author:
Internal investment by Manaaki Whenua—Landcare Research.

### Competing Interests
Thomas R. Etherington is employed by Manaaki Whenua—Landcare Research, and declares that there are no competing interests.

### Author Contributions
- Thomas R. Etherington conceived and designed the experiments, performed the experiments, analyzed the data, prepared figures and/or tables, authored or reviewed drafts of the paper, approved the final draft.

### Data Availability
R scripts to reproduce the examples, virtual experiments, and figures are available as a Supplemental File.

### Supplemental Information
Supplemental information for this article can be found online at http://dx.doi.org/10.7717/peerj.6678#supplemental-information.

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
