# Peer review of "Mahalanobis distances and ecological niche modelling: correcting a chi-squared probability error"

_PeerJ, doi:10.7717/peerj.6678_

## Round 0.1 · original submission · Minor Revisions

Thank for a concise and important paper. Two reviewers, and I, agree this paper is important and nearly ready or acceptance. Please note an important comment from Reviewer 2, which I agree with. Resource selection and niche modeling are not synonymous, though many authors lump these together because of similarities in analysis tools. Please work to revise the sentence in the abstract. There are several other simple fixes required from Reviewer 2.

The anonymous reviewer has several questions worth addressing in the narrative. I leave it to you to answer those and decide if additional modeling is required or if you can address the comments on performance and sample size.

I look forward to your revised submission.

Reviewer 1 ·

Basic reporting

The author proposes a new approach on the use of mahalanobis distance in ecological niche modeling. Niche modeling has been indicated in conservation strategies and there are many methods available and at the same time many uncertainties. This study is important for reducing uncertainties. However, there is no certainty as to which modeling method is most effective and the method consensus approach has been the most indicated. Therefore I thought that in the introduction of this study lacked tacked mentioning what advantages of malalanobis distance compared to other methods in conservation strategies. Can it be used with species of few occurrences? Can it be used with species of restricted distribution?

Experimental design

The experimental delineation of the article has been understood. However, I suggest performing experiments with different numbers of occurrences to evaluate the performance of the method.

Validity of the findings

The article is important so that the uncertainties in the forecasts are reduced. However, it is necessary to reinforce the importance of using the distance malanobis method for niche modeling and what its main impacts are when used erroneously.

Additional comments

I enjoyed learning from the article on improving a modeling method. Many details can make a difference in the parameterization of niche modeling. I thought was very valid. However, I think the study would be better with simulations with different amounts of occurrences. I would like to know the consequences for species conservation.

·

Basic reporting

This paper identifies a pervasive little problem in the use of Mahalanobis distances to characterize the niches of species. The paper is concise and well-written, the bug the author identified is real, and potentially damaging. I believe the paper should be published, almost as it is, with the following minor comments:
1) Figure 2 lacks the first three labels.
2) In the abstract, the author choice of words suggest that niche modeling is equivalent to resource selection, habitat suitability, or distribution modeling. It should be perfectly obvious that modeling a distribution is not synonymous with modeling the selection of resources, for example. Just change the wording so it is clear that Mahalanobis distances are used in all those fields, without implying that the fields are synonymous.

Experimental design

No comment

Validity of the findings

No comment

Additional comments

No comment

---

## Round 0.2 · accepted · Accept

Thank you for your attention to this revision and well crafted and easy to follow response. This was a job well done and congratulations on a strong manuscript.

#

---

## Author Rebuttal · Round 0.2

18 February 2019

Dr. Jason Blackburn
Academic Editor
Dear Jason,

Thank you for your time in organising these reviews, it is very much appreciated. To try and respond to the reviewers' comments with clarity, I have included the reviewers' comments under each heading below in italics, with my responses then following on from that. Also, to make any changes clear, I have also highlighted in yellow additions and corrections that have been made in the revised manuscript. Hopefully you will find this clear to follow, and that you are happy that I have responded sufficiently to the reviewers comments.

**Reviewer 1**

*Basic reporting*

*The author proposes a new approach on the use of mahalanobis distance in ecological niche modeling. Niche modeling has been indicated in conservation strategies and there are many methods available and at the same time many uncertainties. This study is important for reducing uncertainties. However, there is no certainty as to which modeling method is most effective and the method consensus approach has been the most indicated. Therefore I thought that in the introduction of this study lacked tacked mentioning what advantages of malalanobis distance compared to other methods in conservation strategies. Can it be used with species of few occurrences? Can it be used with species of restricted distribution?*

*Experimental design*

*The experimental delineation of the article has been understood. However, I suggest performing experiments with different numbers of occurrences to evaluate the performance of the method.*

*Validity of the findings*

*The article is important so that the uncertainties in the forecasts are reduced. However, it is necessary to reinforce the importance of using the distance malanobis method for niche modeling and what its main impacts are when used erroneously.*

*Comments for the author*

*I enjoyed learning from the article on improving a modeling method. Many details can make a difference in the parameterization of niche modeling. I thought was very valid. However, I think the study would be better with simulations with different amounts of occurrences. I would like to know the consequences for species conservation.*

I would disagree somewhat with Reviewer 1's opening statement that "*The author proposes a new approach on the use of mahalanobis distance in ecological niche modeling.*", as I wouldn't consider

Manaaki Whenua – Landcare Research | P.O. Box 69040, Lincoln 7640, New Zealand
54 Gerald Street, Lincoln 7608, New Zealand
T: + 64 3 321 9999 | F: + 64 3 321 9998 | www.landcareresearch.co.nz

myself to be proposing a new approach, but rather highlighting that many studies and software are applying the established method incorrectly. I think this misunderstanding may be at the centre of some of Reviewer 1's suggestions for additional modelling to assess the impact of sample size – which would be very sensible advice for anyone presenting a new method. I also don't think that it would be appropriate to try and vary sample size within this study, as while the error I am identifying is still present with low sample sizes, it only becomes clear when sample sizes are large. Therefore, I feel that including such additional modelling here would potentially confuse and distract from the focus of the paper that is simply to highlight and explain appropriate use of the chi-squared distribution.

Reviewer 1 also states "*that in the introduction of this study lacked tacked mentioning what advantages of malalanobis distance compared to other methods*". Again, I am slightly confused as I did include the key methodological advantages of the method on lines 55-58 of the original manuscript. However, in thinking about this comment, I am wondering if Reviewer 1 means advantages more in terms of predictive ability in comparison to other methods as Reviewer 1 also states before this that "*there is no certainty as to which modeling method is most effective*". This may well be helpful extra information for the introduction, and there have been studies that have compared the predictive ability of Mahalanobis distances to other methods. Therefore, I have included some additional text and citations highlighted on lines 58-62 to expand on the comparison of the Mahalanobis distance approach to other techniques in terms of predictive ability in the hope that is of help to Reviewer 1.

**Reviewer 2 – Jorge Soberón**

*Basic reporting*

*This paper identifies a pervasive little problem in the use of Mahalanobis distances to characterize the niches of species. The paper is concise and well-written, the bug the author identified is real, and potentially damaging. I believe the paper should be published, almost as it is, with the following minor comments:*

*1) Figure 2 lacks the first three labels.*

*2) In the abstract, the author choice of words suggest that niche modeling is equivalent to resource selection, habitat suitability, or distribution modeling. It should be perfectly obvious that modeling a distribution is not synonymous with modeling the selection of resources, for example. Just change the wording so it is clear that Mahalanobis distances are used in all those fields, without implying that the fields are synonymous.*

Thank you for pointing out both issues. The figure captions have been fixed, and I quite agree with your second point, and with hindsight I haven't worded that sentence well, so I have changed the wording in the abstract to try and make it clear that while Mahalanobis distances are used in all those fields they are not used synonymously.

Kind regards,

Tom Etherington